# Learning-related population dynamics in the auditory thalamus

**Ariel Gilad[1,2]\*, Ido Maor[2], Adi Mizrahi[2,3]**

[1]Department of Medical Neurobiology, Institute for Medical Research Israel Canada, Faculty of Medicine, The Hebrew University, Jerusalem, Israel; [2]The Edmond and Lily Safra Center for Brain Sciences, The Hebrew University of Jerusalem, Jerusalem, Israel; [3]Department of Neurobiology, The Hebrew University of Jerusalem, Jerusalem, Israel

**Abstract** Learning to associate sensory stimuli with a chosen action involves a dynamic interplay between cortical and thalamic circuits. While the cortex has been widely studied in this respect, how the thalamus encodes learning-related information is still largely unknown. We studied learning-related activity in the medial geniculate body (MGB; Auditory thalamus), targeting mainly the dorsal and medial regions. Using fiber photometry, we continuously imaged population calcium dynamics as mice learned a go/no-go auditory discrimination task. The MGB was tuned to frequency and responded to cognitive features like the choice of the mouse within several hundred milliseconds. Encoding of choice in the MGB increased with learning, and was highly correlated with the learning curves of the mice. MGB also encoded motor parameters of the mouse during the task. These results provide evidence that the MGB encodes task- motor- and learning-related information.

**\*For correspondence:**
ariel.gilad@mail.huji.ac.il

**Competing interests:** The authors declare that no competing interests exist.

## Introduction

Learning, the process of acquiring new knowledge through experience, is known to involve disparate brain areas, and particularly the cortex. For example, learning to discriminate between different sensory stimuli leads to changes in the respective primary sensory areas (*Blake et al., 2002*; *Chen et al., 2015*; *Driscoll et al., 2017*; *Gilad and Helmchen, 2020*; *Jurjut et al., 2017*; *Komiyama et al., 2010*; *Li et al., 2008*; *Makino and Komiyama, 2015*; *Poort et al., 2015*; *Yan et al., 2014*). Cortical neural responses have often been shown to strengthen after learning. For example, neural signals in animals that gain expertise in a specific task, show increased activity and result in higher discriminatory power between the learned stimuli (*Gilad et al., 2018*; *Gilad and Helmchen, 2020*; *Poort et al., 2015*; *Wiest et al., 2010*; *Yan et al., 2014*). In other cases, decreased responsiveness to the learned stimuli have been measured. These changes, too, may result in improvement in the discrimination of the stimuli (*Christensen et al., 2019*; *Maor et al., 2020*).

The cortex dynamically interacts with the thalamus via recurrent loops that are thought to involve more than mere sensory processing. Rather, thalamocortical circuitry also encodes high-order processing of cognitive functions such as attention, working memory and learning (*Acsády, 2017*; *Audette et al., 2019*; *Bennett et al., 2019*; *Bolkan et al., 2017*; *Guo et al., 2017*; *McAlonan et al., 2008*; *Rose and Bonhoeffer, 2018*; *Roth et al., 2016*; *Saalmann and Kastner, 2015*; *Schmitt et al., 2017*; *Ward, 2013*; *Williams and Holtmaat, 2019*; *Zhang and Bruno, 2019*). Where do cognitive responses arise and how do they impact basic sensory processing remains an area of active research. In this study, we focused on the medial geniculate body (MGB; auditory thalamus), asking whether and how it represents sensory and cognitive information along learning.

The MGB is, at least in part, a thalamic relay center of the auditory pathway, predominantly receiving direct input from the inferior colliculus (*Calford and Aitkin, 1983*; *Peruzzi et al., 1997*), but also from cortex (*Winer et al., 2001*) and other sources (*Crabtree, 1998*; *Lee, 2015*; *Winer, 1992*). Its projections target the cerebral cortex and numerous other brain areas such as the amygdala (*LeDoux et al., 1991*; *Lee, 2015*; *Tasaka et al., 2020*). Anatomical studies divide the MGB into three main sub-divisions: ventral, dorsal and medial (*Calford and Aitkin, 1983*; *Clerici and Coleman, 1990*; *Hashikawa et al., 1991*; *Imig and Morel, 1985*; *Mo and Sherman, 2019*; *Rouiller et al., 1989*; *Smith et al., 2012*). The ventral MGB relays topographically organized information from the inferior colliculus to the primary auditory cortex (*Rouiller et al., 1989*; *Smith et al., 2012*). This pathway is called the lemniscal pathway, and is considered to be the main auditory processing pathway. While tracing experiment suggest a simple topographical representations of sounds relayed from MGB to cortex (*Hackett et al., 2011*), physiological responses of MGB axon terminals suggest a more complex interaction (*Vasquez-Lopez et al., 2017*). Particularly, the dorsal and medial parts of the MGB are thought to process more complex information than the ventral MGB and not obey strict tonotopy. Dorsal and medial MGB project to higher-order cortical areas (*Huang and Winer, 2000*; *Lee, 2015*; *Tasaka et al., 2020*) and receive cortical feedback, among others, from layer 5 of the auditory cortex (*Bartlett et al., 2000*; *Lee, 2015*; *Llano and Sherman, 2008*). Those parts of the MGB, considered to be part of the non-lemniscal pathway, are well positioned to encode higher-order information of sensory, motor and associative nature, but only few studies have measured these in the auditory thalamus directly (*Jaramillo et al., 2014*).

Learning-related plasticity has been measured in the cortex after training animals to discriminate between two simple sounds, neurons in auditory cortex display several learning related modulations: encoding of higher-order choice information (*Guo et al., 2019*; *Jaramillo et al., 2014*), tonotopic map reorganization (*Maor et al., 2020*; *Polley et al., 2006*), increased responses to a target frequency and decreased responses to a distractor stimulus (*Blake et al., 2002*; *David et al., 2012*; *Ghose, 2004*; *Ohl and Scheich, 2005*). The inferior colliculus, too, has been shown to processes auditory information based on behavioral context and bodily movements, which are also considered high order processing (*Calford and Aitkin, 1983*; *Casseday et al., 2002*; *Gruters and Groh, 2012*; *Yang et al., 2020*). The MGB, being highly interconnected with both downstream and upstream brain regions like the inferior colliculus and cortex (*Tasaka et al., 2020*), is expected to encode learning-related activity. Indeed, the MGB has been shown to encode the choice of the mouse during an auditory discrimination task (*Chen et al., 2019*; *Jaramillo et al., 2014*). Furthermore, a large body of evidence indicates that the medial MGB is involved in auditory fear conditioning, by providing fast, less refined auditory information to the lateral amygdala (*Han et al., 2008*; *Herry and Johansen, 2014*; *Maren and Quirk, 2004*; *Quirk et al., 1995*; *Romanski and LeDoux, 1992*; *Weinberger, 2011*). To study learning related modulations in the MGB in the context of perceptual learning, we chronically imaged MGB population responses to sounds as mice learned a go/no-go auditory discrimination task. We find learning-related modulations in the MGB, and a particularly strong modulation to choice signals.

## Results

### Calcium imaging from the MGB along learning using fiber photometry

To study learning-related changes in the MGB, we injected AAV-GCaMP6f into the MGB of C57BL/6 mice and implanted a 400 μm optical fiber directly above the injection site. After a week of handling and habituation to head-fixation, we trained mice on a go/no-go auditory discrimination task. Each trial started with a visual start cue (orange LED; duration 0.1 s; 2 s before stimulus onset) followed by an auditory stimulus, either a go or a no-go pure tone sound (*Figure 1A*; duration 1 s; sounds separated by 0.5 octave; mostly 10 kHz for go and 7.1 kHz for no-go; see Materials and methods). After stimulus offset, we counted licks in a virtual response window of 3 s. Licking in response to the go sound were counted as 'hit' trials and rewarded with a drop of water. Withholding licking in response to the no-go sound were counted as correct rejection trials (CR), and were not punished or reinforced. Licking in response to the no-go sound were counted as false alarm trials (FA) that were followed by a mild punishment in the form of white noise (duration 3 s). Withholding licking for the go sound were counted as Miss trials, and were not punished. As mice learned to discriminate

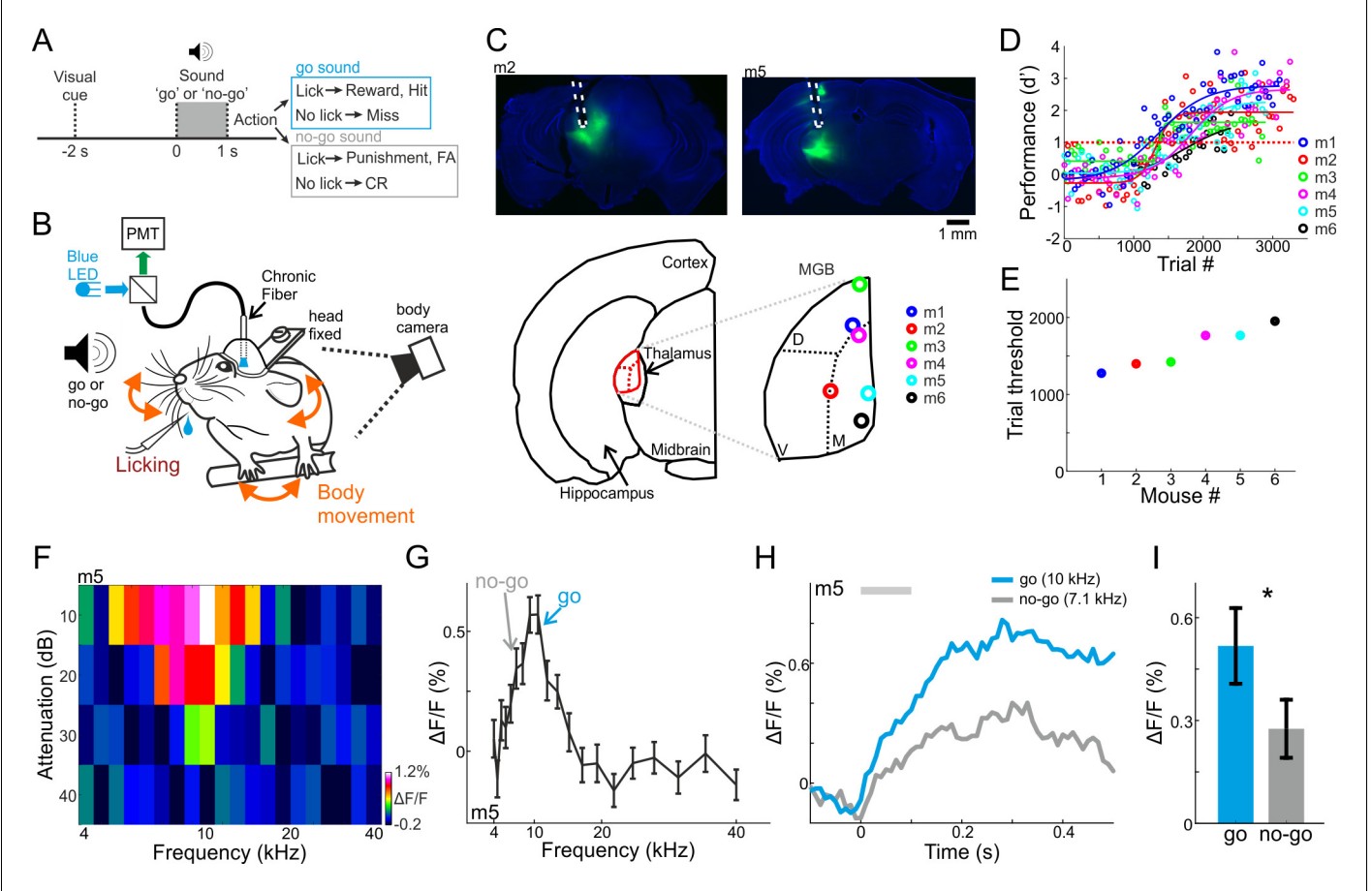

**Figure 1.** Behavioral paradigm, performance, and frequency tuning. (A) Trial structure of a go/no-go auditory discrimination task and possible trial outcomes. (B) Behavioral setup for head fixed behaving mice along with simultaneous fiber photometry in the medial geniculate body (MGB). (C) *Top:* Fluorescent images of two coronal slices from two different mice, showing GCaMP6f (green) in the MGB along with the fiber track highlighted in white. DAPI staining in blue. *Bottom:* Localization of the fiber tip in the MGB for all six mice. (D) Behavioral learning curves for all mice (n = 6) depicting the performance (d') as a function of trial number. Each learning curve was fitted with a sigmoid function. Dashed red line indicates the performance threshold (d'=1). (E) Learning threshold (the trial number where the learning curve crossed the performance threshold) for all mice. (F) A frequency response area plot (attenuation versus frequency) from one example recording of one example mouse. (G) Frequency tuning curve for the same example mouse. The frequencies used as go and no-go are marked by arrows. (H) Average responses to the go (blue) and no-go (gray) sounds. (I) Average evoked response (calculated from the 100 ms after stimulus onset; gray bar in 'H') to the go and no-go sounds from all mice. Error bars are s.e. m across mice. *p<0.05. Wilcoxon sign rank test.

The online version of this article includes the following figure supplement(s) for figure 1:

**Figure supplement 1.** Behavioral performance and learning curves.

**Figure supplement 2.** Frequency tuning for each mouse.

between the two sounds we continuously imaged population responses in the MGB in addition to monitoring their body movements during the task (*Figure 1B*). We first imaged six mice across learning. The fiber tips of the six mice were reconstructed to the higher-order regions of the MGB (*Figure 1C*, medial and dorsal parts grouped as MGB for simplicity). Mice learned the task within 1200–2000 trials as assessed by d' (defined as d'=Z(Hit/(Hit+Miss)) – Z(FA/(FA+CR) where Z denotes the inverse of the cumulative distribution function; performance threshold was defined as d'=1; *Figure 1D,E*; fitted with a sigmoid function (*Bathellier et al., 2013*; *Maor et al., 2020*). The learning curves with respect to the go and no-go trials shows that individual mice varied widely in performance and strategy (*Figure 1—figure supplement 1*). Learning the task took different forms: some mice increased their CR rate, others had a steep increase in hit rate, whereas others gradually increased both hit and CR rates. Thus, mice learned the task using a range of behavioral strategies.

Prior to training we imaged fluorescent fiber response to a range of frequencies while the mouse was passively listening in the awake state (4–40 kHz; 10, 20, 30, 40 dB attenuations relative to 62 dBSPL; duration 0.1 s; see Materials and methods). Population responses in the MGB displayed frequency responses, biased to the low frequency range (*Figure 1F*). In the six mice shown below, the response to the go frequency was significantly higher as compared to the no-go frequency (*Figure 1G–H*, one example mouse; *Figure 1I*, average of all six mice; p<0.05; Signed rank test; *Figure 1—figure supplement 2A* for the tuning curves of all mice).

The tuning curves of individual neurons in the dorsal and medial parts of the MGB are generally broader than ventral MGB (*Bordi and LeDoux, 1994*; *Weinberger, 2011*), and they represent multiple frequencies (*Hackett et al., 2011*). It is therefore surprising that the responses we measured were rather sharply tuned. Further surprising was that our signals peaked at lower frequencies, while the bulk signal of photometry is summing activity from multiple neurons. Whereas we cannot determine the source of the signal explicitly, the fiber tip locations showed that most recordings were from the dorsal and medial MGB. The full frequency response areas showed that our signal spanned 2.5 octaves, at least in high intensities. However, we cannot exclude that feedback connections from higher cortex (*Vasquez-Lopez et al., 2017*) or signal leaking from the ventral MGB contributed to the signal we recorded. Thus, the main source of the fluorescent signal is primarily, though not exclusively, of non-leminsical origin.

## MGB encodes the choice of the mouse

To evaluate how sounds and other task attributes were represented in the MGB across learning, we plotted responses to the go and no-go sounds as mice learned the task. *Figure 2A* shows responses to the go and no-go sounds of one representative mouse (*Figure 2A*). The most evident change in MGB responses across learning was during the late part of the trial (0.6–1 s after stimulus onset), and particularly so for go trials (*Figure 2A*).

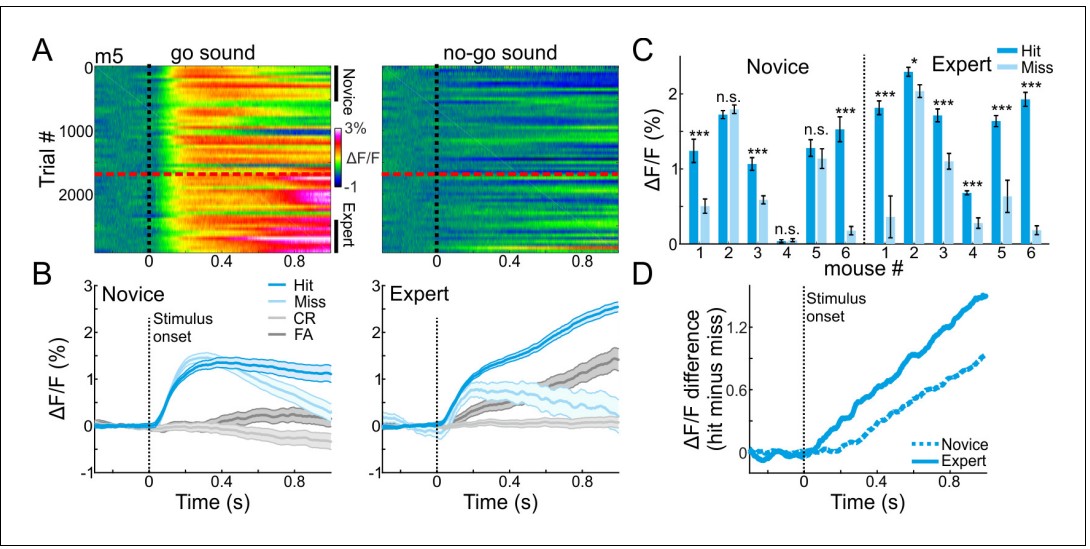

**Figure 2.** MGB encodes choice in expert mice. (**A**) 2-dimensional plots of the calcium responses in MGB during the trial (x-axis) and across learning (y-axis; 50 trial bins) for one example mouse divided into go (left) and no-go (right) sounds. Dashed black line indicates stimulus onset (1 s duration) and dashed red line indicates the learning threshold. (**B**) Calcium responses when the mouse was novice (left) and expert (right). Traces are shown separately for different trial types (hit, miss, CR and FA trials; same data as in 'A'). Shaded error bars are s.e.m across trials. (**C**) Mean calcium response during stimulus presentation per mouse in hit and miss trials when mice were novice (left) and expert (right). Error bars are s.e.m across trials. (**D**) Choice responses (defined as the response difference between hit and miss trials) during the trial, averaged across all mice when they were novice (dashed line) and expert (solid line). *p<0.05. ***p<0.001. n.s. – not significant. Wilcoxon rank sum test.

The online version of this article includes the following figure supplement(s) for figure 2:

**Figure supplement 1.** Calcium responses in different trial types for all mice.

A particularly informative comparison is between hit and miss trials, where the stimulus is identical but the choice of the mouse is different (either lick or no-lick). Thus, differences between hits and misses represent encoding of choice. In expert mice (defined as the last 500 trials), MGB responses were higher for hit as compared to miss trials, and more so than in the novice mice (defined as the first 500 trials; *Figure 2B*; compare blue to light blue traces; For comparison, CR and FA trials are plotted in gray; MGB responses for all six mice individually are presented in *Figure 2—figure supplement 1*). Higher responses in hit versus miss trials were evident in all (6/6) expert mice and in 50% (3/6) of novice mice (*Figure 2C*; p<0.05; Wilcoxon rank sum test for each mouse separately; minimum 40 trials for hit or miss). Choice responses (i.e. hit minus miss) increased gradually after stimulus onset, and were stronger in expert mice (*Figure 2D*). These data suggests that the MGB encodes more than only sounds. MGB encodes the choice of the mouse, which implies that the auditory thalamus is involved in higher-level sensory-motor processing or some cognitive attributes of the task.

## MGB encodes sounds early and choices late

To further explore the role of MGB in high level processing versus low-level sound processing, we tested the discrimination between sensory stimuli and choices at the single trial level. To do so, we calculated the receiver operating curve (ROC) and derived the area under the curve (AUC) between pairs of different trial-type distributions. We define two different AUC measures by comparing hit trials to either FA (stim AUC; magenta; that is different stimuli but similar choice) or to Miss (choice AUC; green; that is similar stimuli but different choice) trials (*Figure 3A*). The AUC value ranges from 0 to 1 and quantifies the accuracy of an ideal observer. AUC values close to 0.5 indicate low discrimination whereas values away from 0.5 indicate high discrimination. Together, these comparisons encompass the full breadth of options in the task (see below for additional trial types). In

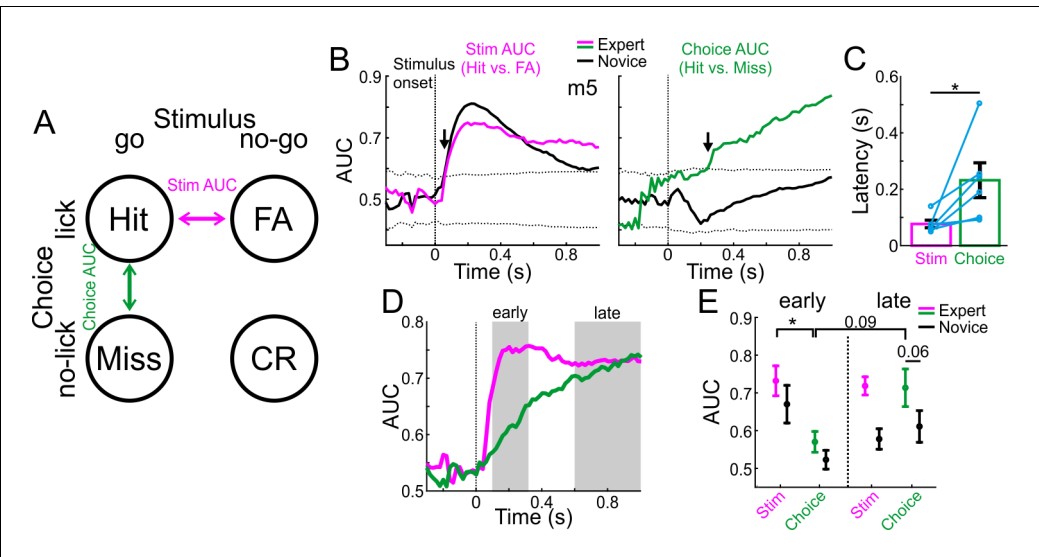

**Figure 3.** MGB discriminates early between stimuli and late between choices. (A) Schematic of the two single trial discrimination measures: hit vs. FA (magenta; Stim AUC) or hit vs. miss (green; Choice AUC). (B) Stim and Choice AUCs during the trial for one example mouse during novice (black lines) and expert (colored lines). Dashed black lines display the mean ±3 s.t.d of AUC from trial shuffled data. Arrows indicate the latency of the AUC measure (i.e. the first time point the signal exceeded the shuffled data). (C) Latency of discrimination for the Stim and Choice AUCs for all mice (averaged and marked individually). Error bars are s.e.m across mice (n=6). (D) Stim and Choice AUCs averaged across all expert mice. Early and late times are marked in gray. (E) The two AUC measures averaged during early (left) and late (right) times when mice were novice (black lines) and expert (colored). Error bars are s.e.m across mice. *p<0.05. Wilcoxon signed-rank test.

The online version of this article includes the following figure supplement(s) for figure 3:

**Figure supplement 1.** Choice-AUC responses for all mice.

**Figure supplement 2.** Stim-AUC and Choice-AUC based on other trial type pairs.

addition, we calculated AUC values for each time frame along the trial for the novice and expert conditions separately (i.e. first and last 500 trials). As a control, we calculated the sample distribution of trial-shuffled data and calculated the AUC identically. We define significance as the time point when the observed AUC values exceed ±3 std above the mean of the shuffled distribution (dashed lines in *Figure 3B*). For example, AUC data from one mouse is shown in *Figure 3B*. The 'stim-AUC' value is significantly discriminative early in the trial in both the novice and expert cases (*Figure 3B*, <100 ms from stimulus onset; arrows). This example shows that MGB responses carry highly discriminative information between the stimuli early in the trial and that this does not change following learning. In contrast, 'choice-AUC' reached significance only in the expert condition and only later in the trial (*Figure 3B*, green trace,>200 ms from stimulus onset; arrow).

To quantify this effect across mice, we defined the exact latency for discrimination as the time it takes to reach significance for each AUC measure in the expert case (arrows in *Figure 3B*). The latency of discrimination for 'stim-AUC' was significantly earlier than that of the 'choice-AUC' (*Figure 3C*; p<0.05; Signed rank test; n = 6 mice). In addition, the onset of the choice-AUC was significantly earlier than the onset of licking in expert mice, implying that choice signal in the MGB precedes the execution of the motor command (p<0.05; Signed Rank test; n = 6 mice; Onset defined in both traces as the peak of the 2$^{nd}$ derivative; Choice onset = 207 ± 109 ms; Lick onset = 385 ± 130 ms; mean ±std). Averaged across all mice, stim-AUCs increased early and abruptly whereas choice-AUC increased gradually (*Figure 3D*; choice-AUCs for individual mice are shown separately in *Figure 3—figure supplement 1*). We then pooled together the AUC values during the early (0.1–0.3 s after stimulus onset) and late (0.6–1 s after stimulus onset) times of the trial (*Figure 3E*). We found that 1) Stim-AUC was significantly higher than choice-AUC early in the trial (p<0.05; Signed rank test; n = 6 mice), 2) Choice-AUC was higher, yet not significantly, during the late part of the trial compared to early times (p=0.09; Signed rank test; n = 6 mice), and 3) Choice-AUC was higher, yet not significantly, in the expert compared to the novice state, particularly during late times during the trial (*Figure 3E*; p=0.06; Signed rank test; n = 6 mice). In addition, we calculated choice and stim AUCs based on other pairs of trial types: CR vs. Miss for an alternative stim-AUC and FA vs. CR for an alternative choice-AUC (*Figure 3—figure supplement 2*). The alternative stim-AUC displayed similar early onset discrimination in both novice and expert cases. In contrast, the choice-AUC showed significant discrimination only in some mice, but not all. This was mainly due to the fact that the responses were quite weak to the no-go sound (see *Figure 2—figure supplement 1* for ΔF/F responses in CR and FA trials), resulting in low discrimination values. In summary, these data indicate that stimulus information is present in the MGB relatively early, whereas choice information develops relatively late - several hundred milliseconds after stimulus onset.

## Plasticity in MGB correlates with learning

Continuous imaging throughout the whole learning process enabled us to investigate the activity of the MGB on a trial-by-trial manner, and compare it to behavioral performance of each mouse. For this analysis, which was focused on go trials only, we averaged the calcium signals at late times during the trial (0.6–1 s after stimulus onset) to obtain MGB responses as a function of learning (termed 'MGB response curve'). The MGB response curve was sigmoid-like, resembling the learning curve of the mouse (*Figure 4A* shows data from two mice; line is a sigmoid function fit). Next, we calculated the MGB response curve for each time point along the trial separately, correlated it with the learning curve, and plotted the correlation coefficient along time (see Materials and methods). Positive correlations depict high similarity between the MGB response curve to the learning curve at the given time point. The correlation increased after stimulus onset, indicating a strong relationship between MGB modulations and the learning process (*Figure 4B* - two representative mice; *Figure 4C*- all mice). To find the maximum point of change for each MGB response curve, we normalized sigmoid fits of the MGB response curves for each mouse separately, and derived the trial threshold for each curve (*Figure 4D*; defined as the trial number in which the normalized curve crosses 0.5). The trial threshold of the MGB curves matched nicely the learning threshold of each mouse (*Figure 4E*; r = 0.89; p<0.05). Calculating the learning curves based on earlier times (0–0.3 s after stimulus onset) resulted in MGB curves which are much less sigmoid-like and their threshold did not match the learning curves of each mouse (r = 0.3; p>0.05), indicating that learning dynamics develop relatively late. To test whether MGB responses had undergone changes not directly related to learning, we compared between the responses to tones in passively listening mice before and after mice learned the

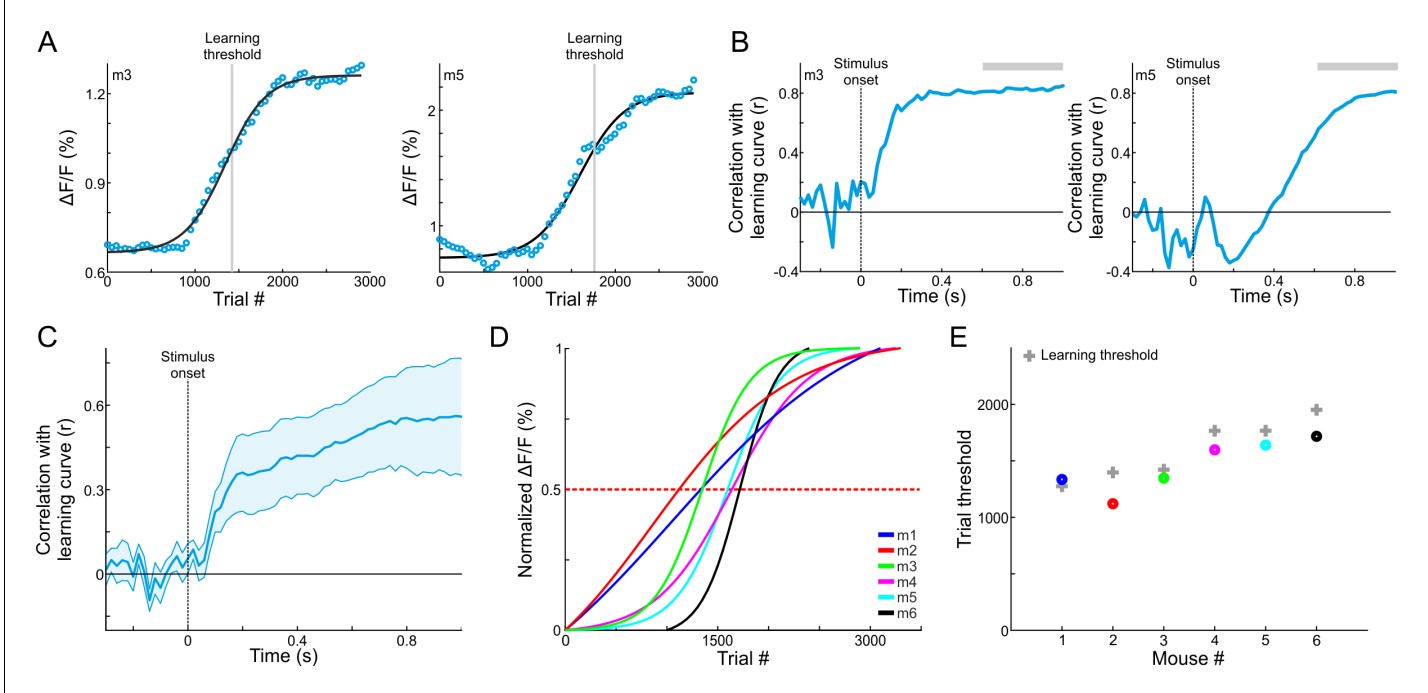

**Figure 4.** MGB changes correspond with gaining expertise. (A) MGB responses (averaged during late times, 0.6–1 s after stimulus onset; gray bars in B) as a function of learning in two example mice. Each MGB response curve was fitted with a sigmoid function (black lines). The learning thresholds are marked with a gray line. (B) Correlation coefficient as a function of time between the MGB response curve and the learning curve of the two example mice shown in 'A'. Positive values indicate that MGB responses change in a similar way to the behavioral performance of the mouse. (C) Same as B, but averaged across all six mice. Error bars are s.e.m across mice. (D) Normalized fits of the MGB response curve for all mice. Dashed red line indicates threshold. (E) Maximal change in MBG responses (i.e. the trial number in which the MGB fit crossed 0.5, dots) superimposed on the learning thresholds (gray crosses).

The online version of this article includes the following figure supplement(s) for figure 4:

**Figure supplement 1.** Frequency tuning is similar before and after learning.

task. We found no differences between the MGB frequency tuning curves or the response to the go frequency during passive listening, before and after learning (*Figure 4—figure supplement 1*), indicating that MGB plasticity occurs specifically for task related signals, at least at the population level. In summary, we found that changes in MGB responses are positively correlated with learning as evident in similar modulation of the choice signal and the performance of individual mice.

## Body movements affect MGB responses

The strong relationship between MGB plasticity and learning prompted us to further investigate other parameters that may affect learning-related modulations, specifically related to encoding of choice. Factors that could affect MGB responses are parameters related to movement (e.g. forelimb movement, nose twitching, whisking, licking etc.), which were recently found to have substantial impact on neuronal activity in different brain areas (*Gilad et al., 2018*; *Gilad and Helmchen, 2020*; *Musall et al., 2019*; *Stringer et al., 2019*). As mentioned above, we continuously monitored the body movements of the mouse throughout the experiment (*Figures 1A* and *5A*). As expected, changes in body movements were correlated with learning. Around the time mice crossed the learning threshold, they exhibited more body movements as evident by plotting their movement probability (*Figure 5A*, right). Movement probability was high both within the stimulus period (0–1 s after stimulus onset) and, as expected, during the response period (*Figure 5A,B*). During pre-stimulus periods (−1 to 0 relative to stimulus onset) expert mice moved significantly less compared to the novice case, indicating a higher level of alertness (p<0.01; Signed rank test). Movement probability during stimulus presentation across learning was sigmoid-like, and significantly higher in experts compared to novice mice (*Figure 5C,D*; p<0.05; Signed rank test; n = 6 mice). Thus, as part of the

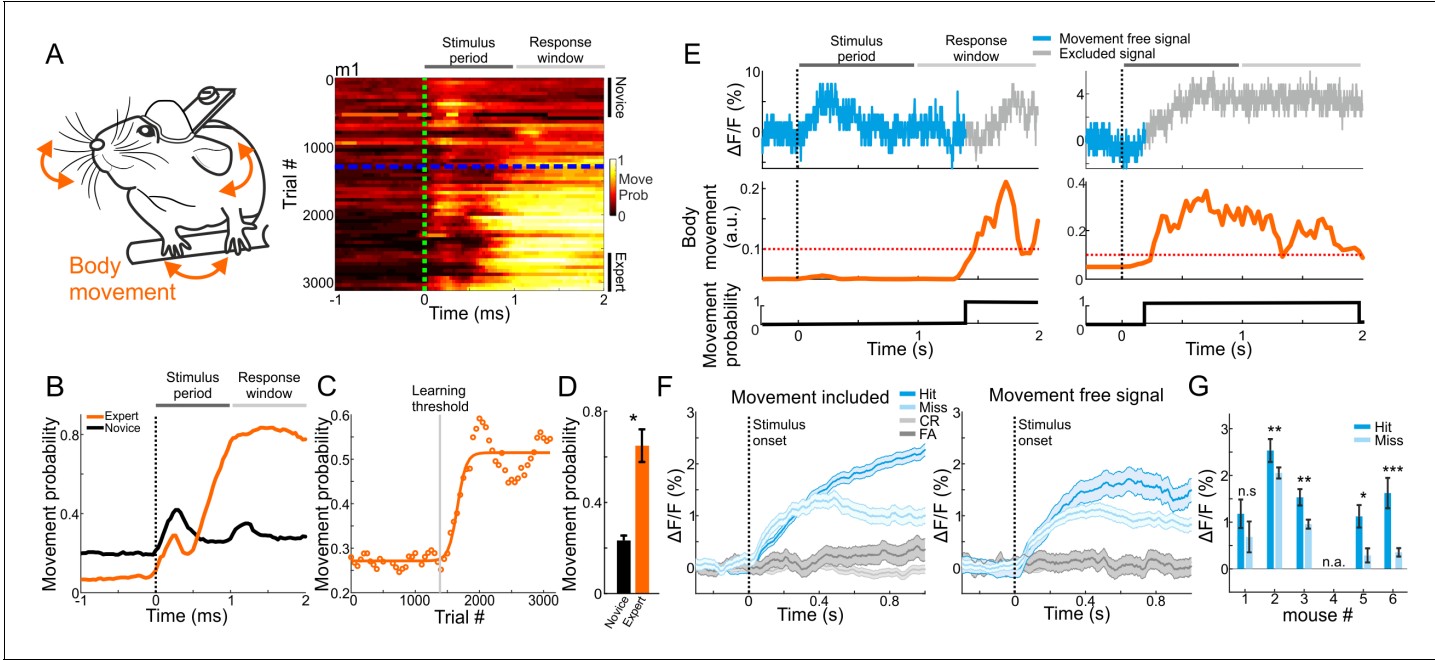

**Figure 5.** Body movements affect MGB responses. (**A**) Left: schematic illustration of mouse body movements during the task. Right: A 2-dimensional plot of movement probability during go trials within the trial (x-axis) and across learning (y-axis; 50 trial bins) for an example mouse. Dashed green line indicates stimulus onset (1 s duration) and dashed blue line indicates learning threshold. Color scale bar indicates min/max of movement probability. (**B**) Movement probability during the trial for the example mouse in A when it was novice (black) and expert (orange). (**C**) Movement probability (averaged during late times; 0.6–1 s after stimulus onset; go trials) across learning fitted with a sigmoid function. Same data as in 'B'. (**D**) Movement probability during late times for novice and expert averaged across all six mice. Error bars are s.e.m across mice. *p<0.05. Signed-rank test. (**E**) Single trial traces of MGB calcium response (top), body movement (middle) and a binary movement vector (bottom; thresholding the body movement, dashed red line; Materials and methods). To exclude effects of body movements, MGB responses were truncated just before the movement onset on a trial-by-trial basis (gray traces in each trial). (**F**) MGB calcium responses for different trial types when including (left) and excluding (right) body movements. Data is from one example mouse (same as *Figure 2B*). Error bars are s.e.m across trials. (**G**) Mean MGB responses (averaged during stimulus presentation) for movement-free hit and miss trials per mouse. Data from expert mice. Error bars are s.e.m across trials. *p<0.05. **p<0.01. ***p<0.001. n.s. – not significant. n.a. – not available. Wilcoxon rank sum test.

The online version of this article includes the following figure supplement(s) for figure 5:

**Figure supplement 1.** Movement-free MGB choice responses along learning.

**Figure supplement 2.** Movement of different body parts with respect to stimulus onset.

development of choice responses in MGB, the body movements of mice during the task were also strongly related to the learning process.

The strong effects of movement suggest that the abovementioned MGB 'choice' signal may simply represent body movements rather than choice, per se. Indeed, some MGB responses and body movement covaried strongly whereas others did not (*Figure 5E*). Thus, to rule out the possibility that movement is the dominant feature in our calcium signal, we calculated 'movement free' MGB responses by detecting the movement onset in each trial and truncating the MGB signal when mice moved (gray traces in *Figure 5E*; see Materials and methods). This allowed us to evaluate separately MGB responses without the body movements. Movement-free MGB responses still encoded the choice of the mouse. Specifically, we found higher responses in hit compared to miss trials (*Figure 5F*). Notably, the effect size was smaller and levels of statistical significance were weaker when compared to the signal with movement (*Figure 5G*; compare to *Figure 2C* 'Expert'). Importantly, when mice were still novice, movement-free MGB responses did not encode choice, indicating that choice encoding develops with learning, independent of movement (*Figure 5—figure supplement 1A*). Indeed, movement-free MGB responses were also enhanced as a function of learning, indicating that enhanced responses in MGB are related to a learning processes in ways that are not only directly linked to the movements of the mouse (*Figure 5—figure supplement 1B,C*). To

rule out movement-related artifacts, we also imaged mice during the task while exciting the MGB with a wavelength that does not excite the calcium indicator (565 nm). This non-calcium signal was analyzed in a similar manner to the calcium signal and resulted in a relatively flat trace throughout the trial, showing that there are no movement related artifacts (data not shown). In summary, body movements during the task are evident in the calcium signal and are learning-related. Nevertheless, MGB responses still maintains choice information that is separate from the motor parameters and, critically, develops with learning. These data strengthen the claim that MGB encodes high-level information.

## Learning-related changes in MGB differ in different frequency bands

As noted above, MGB responses were tuned to frequency (*Figure 1F–I*). We next tested whether the changes we observed in MGB are general or specific to the response properties at the range of frequencies we used. To this end, we trained three additional mice on the task. In these mice, the go frequency did not correspond to the best frequency. That is the underlying MGB population response was not highest for the target frequency (tuned away from the go frequency; *Figure 6A* shows one mouse with the go signal in the outskirts of the tuning curve; *Figure 1—figure supplement 2B* for the tuning curves of all three mice). Interestingly, MGB responses in all three mice still showed choice encoding but the absolute direction of change was in the opposite direction as compared to the first cohort of mice (compare *Figure 6B* to *Figure 2C*). Specifically, the MGB responses to the hit trials were now significantly lower than to the miss trials and this was evident in expert mice only (*Figure 6B*; p<0.001; Signed rank test for each mouse separately; minimum 24 trials for hit or miss). In expert mice, choice encoding gradually developed with time along the trial and was evident only in expert mice (*Figure 6C*). Here again, MGB response curves were sigmoid-like (*Figure 6D*) and strongly correlated with learning (*Figure 6E*). A more detailed description of these effects is shown in *Figure 6—figure supplement 1*. Taken together, we infer that learning enhances responses to the target frequency while simultaneously suppressing activity away from the target frequency. Notably, both phenomena still result in the same effect (i.e. encoding of choice), but likely exploit different mechanisms.

## Discussion

We describe learning-related changes in evoked responses to sounds in the auditory thalamus. We find that auditory thalamus encodes the choice of the mouse several hundred milliseconds (~200 ms; *Figure 3C*) after sound onset, by specifically enhancing activity to the go frequency as learning proceeds. Thus, late thalamic responses to the same sound are different depending on its learned state.

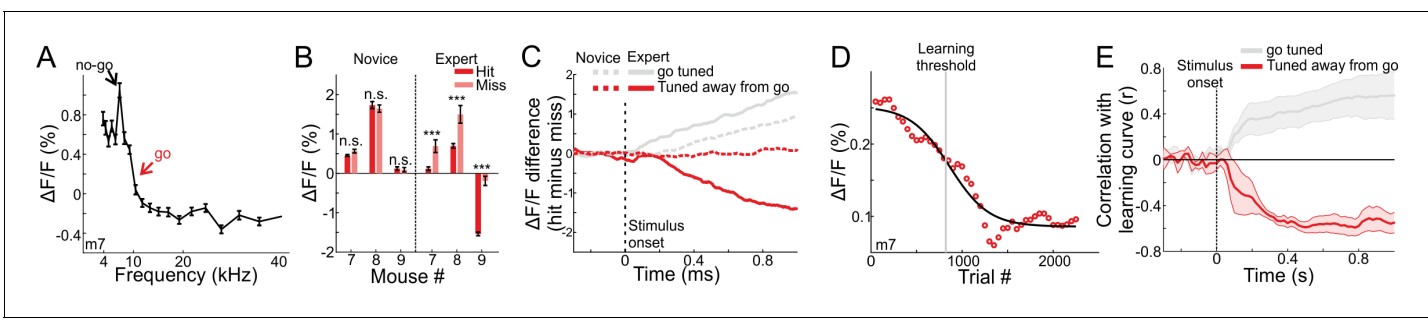

**Figure 6.** MGB responses tuned away from the go frequency are suppressed during learning. (**A**) Frequency tuning curve for an example mouse with peak tuning frequency away from the go sound. (**B**) Mean MGB calcium response during stimulus presentation per mouse in hit and miss trials. Left- novice mice, Right- expert mice. Error bars are s.e.m across trials. \*\*\*p<0.001. n.s. – not significant. Wilcoxon rank sum test. (**C**) Choice responses, averaged across the three mice that are tuned away from the go frequency, for the expert (solid line) and novice (dashed line) conditions. Gray traces are for the go-tuned mice, same as *Figure 2D*. (**D**) MGB response curve along learning in one mouse that is tuned away from the go frequency (compare to *Figure 4A*). (**E**) Correlation between MGB response curves and learning curves as a function of time, averaged across the three mice. Error bars are s.e.m across mice. Gray traces are for the go-tuned mice, same as *Figure 4C*.

The online version of this article includes the following figure supplement(s) for figure 6:

**Figure supplement 1.** MGB responses tuned away from the go frequency are suppressed during learning.

MGB responses alone allowed us to predict with high precision when each mouse learned the task. Taken together, we provide strong evidence relating MGB plasticity to learning a simple auditory discrimination task.

Higher-order thalamic nuclei are part of a thalamocortical loop that are suggested to encode high-order cognitive information (*Bolkan et al., 2017*; *Guo et al., 2017*; *Schmitt et al., 2017*). Choice encoding in the thalamus (*Chen et al., 2019*; *Gimenez et al., 2015*; *Jaramillo et al., 2014*) bears similarity to the observations made in the cortex (*Gilad et al., 2018*; *Guo et al., 2019*; *Harvey et al., 2012*; *Pho et al., 2018*; *Siegel et al., 2015*; *Yang et al., 2016*), indicating the intimate involvement of thalamocortical loops during learning. The appearance of learning related choice encoding was consistent across mice and evident at the population level (*Figure 3*). Since we are unable to dissect the fiber photometry signal to its single cell components, we can only speculate about the nature of cells encoding the choice. The increase in choice signal could arise from small contributions of a large fraction of MGB neurons or from large contributions of a smaller population. A previous study found that neurons in the MGB (and similarly in cortex) mediate choice but to a limited extent, whereas only a small fraction (16%) of recorded neurons encoded choice information, supporting the latter possibility (*Jaramillo et al., 2014*). One noted difference is that *Jaramillo et al., 2014* recorded responses from all MGB sub-nuclei, including ventral MGB, while our measurements were more localized to the medial and dorsal MGB. Ventral MGB might not encode choice, and thus previous work may have underestimated the contribution of single neurons in the non-lemniscal pathway in signaling choice. In fact, further functional dissection is warranted. For example, dorsal MGB will likely be affected more prominently by motor parameters compared to the medial MGB (*Bartlett et al., 2000*; *Huang and Winer, 2000*; *Lee, 2015*). Further, specific populations of MGB neurons that project to different targets will likely transfer specific learning-related information in a selective manner *Chen et al., 2019*; as has been shown for the enhancement of choice information in the cortico-striatal pathway (*Guo et al., 2019*). Simultaneous recordings, preferably with electrophysiology, with tight regulation of the information transfer between thalamus and cortex will be instrumental in deciphering the origin of choice information.

The observed changes in MGB points to pronounced plasticity during the training phase. The mechanisms of this plasticity are unknown and may involve several factors such as inhibitory effects, changes in excitation-inhibition balance, and other forms of synaptic plasticity in specific pathways to the MBG (*Audette et al., 2019*; *McAlonan et al., 2008*; *Rose and Bonhoeffer, 2018*). Learning-related dynamics have been shown to involve disparate circuit elements such as deep cortical layers (*Audette et al., 2019*; *Olsen et al., 2012*), specific inhibitory neuron subtypes (*Hashikawa et al., 1991*; *Khan et al., 2018*; *Peruzzi et al., 1997*) or other thalamic sub-nuclei (*Audette et al., 2019*; *Bennett et al., 2019*). Given that cortex encodes choice, higher-order thalamocortical connections are good candidates to drive synaptic plasticity in cortex during learning (*Audette et al., 2019*). It is unknown whether the synaptic changes from the thalamus to the cortex are the source of change in the MGB or whether corticothalamic feedback is the source. The prolonged latency of choice development in our study, implies an involvement of a late component that may suggest top-down feedback interaction. In addition, an interesting sub-thalamic nucleus that may be involved in gating higher-order information is the thalamic reticular nucleus (TRN) which sends GABAergic projections to MGB (*Crabtree, 1998*; *Lee, 2015*). TRN could gate incoming sensory information, where in the novice case it is expected to exert strong inhibition to MGB which is then decreased with learning; thus enabling enhanced activation to the go sound already at the level of the MGB (*McAlonan et al., 2008*; *Nakajima et al., 2019*). Such a mechanism may contribute to the enhanced discrimination between go tuned and no-go tuned recordings sites in the MGB of expert mice. Selective disinhibition or inhibition could enhance go-tuned neuronal populations or suppress no-go tuned populations, respectively.

Local and global functional connectivity are rich in cortical networks, including auditory cortex (*Rothschild et al., 2010*; *Maor et al., 2016*). Selective changes in local and global functional connectivity based on task demands is not unprecedented, and the mechanisms underlying it has been attributed to the thalamus (*Nakajima and Halassa, 2017*). We show that as mice learn the task, the target frequency enhances activity in tuned population whereas non-tuned populations are simultaneously suppressed. This phenomenon may make the target stimulus more salient based on context and bear similarities to other enhancement/suppression mechanisms found in visual thalamus (*Fisher et al., 2017*; *Jones et al., 2012*; *Nakajima et al., 2019*; *Wilke et al., 2009*). Comparing

the different MGB subdivisions, we hypothesize that that medial and dorsal MGB neurons encoding non-target frequencies will be suppressed, whereas the ventral MGB may continue with a similar response profile for all frequencies. Since higher-order MGB projects to higher-order auditory cortex (part of the non-lemniscal), this information may bypass incoming information from the lemniscal pathway and may aid in increasing discrimination performance.

We find that motor parameters, such as body movements, are also learning-related. As mice learn to associate between the go sound and a reward, they increase their body movement in preparation for licking and the upcoming reward, well within the stimulation period. MGB responses were also affected by body movements to different extents, as might be expected by the motor efferent cortical feedback from layer 5 (*Bartlett et al., 2000*; *Lee, 2015*; *Llano and Sherman, 2008*). Another possibility is that movement related encoding may ascend from the inferior colliculus, an upstream area (*Gruters and Groh, 2012*; *Yang et al., 2020*). The fact that MGB still encodes choice after truncating movement signals, indicates that the MGB does not inherit all task information from the inferior colliculus, but rather integrates information coming from both top-down and bottom-up inputs. The effect of motor parameters on neuronal signals were recently observed in a brain-wide manner, emphasizing the need to strictly monitor and consider motor parameters during different tasks (*Gilad et al., 2018*; *Gilad and Helmchen, 2020*; *Musall et al., 2019*; *Stringer et al., 2019*); and particularly so when measuring higher-order information. In summary, we show that the auditory thalamus encodes high-order information during the time course of learning, implying that the learning process requires brain-wide activity spanning across both cortical and subcoritcal regions.

## Materials and methods

### Key resources table

| Reagent type (species) or resource | Designation | Source or reference | Identifiers | Additional information |
|---|---|---|---|---|
| Transfected construct (Adeno-associated virus) | pAAV.Syn.GCaMP6f. WPRE.SV40 | Addgene | RRID:Addgene_100837 | |
| Software, algorithm | Matlab, Labview | Mathworks, National Instruments | | |

### Animals

A total of n = 9, 8–16 week-old female C57BL/6 mice were used in this work. All experiments were approved by Institutional Animal Care and Use Committee (IACUC) at the Hebrew University of Jerusalem, Israel (Permit Number: NS-19-15706-4).

### Surgery

To express a calcium indicator in MGB neurons and implant an optical fiber for imaging, mice were anesthetized with 2% isoflurane (in pure $O_2$) and body temperature was maintained at 37°C. We applied local anesthesia to the area of surgery (lidocaine 1%), exposed and cleaned the skull. Next, we drilled a small hole in the skull and injected 240 nl of an AAV virus pAAV.Syn.GCaMP6f.WPRE. SV40 (AAV9; from Addgene) using a glass pipette, to the dorsal or medial part of the MGB (3.3 mm posterior to bregma; 1.9 mm lateral to bregma; 3 mm in depth). Using the same coordinates, we then inserted a 400 μm optical fiber with an attached cannula (CFMC14L05; Thorlabs), directly above the injection site (2.95 mm deep). We chronically fixed the fiber position using dental cement that was mixed with a few drops of superglue for extra strength and stability. Finally, a metal post for head fixation was glued onto the bone in the back side of the right hemisphere. This procedure enabled a chronic and reliable imaging of population calcium responses from the MGB.

### Auditory discrimination task

Mice were trained on a go/no-go auditory discrimination task (*Figure 1A*). We used the exact same hardware and software of an automated behavior system we recently described called the "Educage" (see details in *Maor et al., 2020*). Unlike the Educage where mice are free to train from their home, here, we trained mice on this system while head fixed in an isolated acoustic chamber. In short, each trial started with a visual start cue (orange LED placed in front of the mouse; duration 0.1

s; 2 s before stimulus onset) followed by an auditory stimulus, either a go or a no-go pure tone (*Figure 1A*; duration 1 s; 62 dBSPL; sounds separated by 0.5 octave). All mice were trained on 10 kHz for go and 7.1 kHz for no-go except for mouse #3 which was trained on 8.2 kHz for go and 5.8 kHz for the no-go. After stimulus offset, we counted licks in a virtual response window of 3 s. Licking in response to the go sound were counted as 'hit' trials and rewarded with a drop of water. Withholding licking in response to the no-go sound were counted as correct rejection trials (CR), and were not punished or reinforced. Licking in response to the no-go sound were false alarm trials (FA) that were followed by a mild punishment of white noise (duration of 3 s). Withholding licking for the go sound were counted as misses, and were not punished. The licking detector remained in a fixed and reachable position throughout the entire trial and mice were free to lick at any time. Licking before the response cue was allowed and did not lead to punishment or early reward. Note that the visual cue merely signals the start of the trial, but had no predictive power with respect to go or no-go condition.

## Training and performance

Nine mice were trained on the task. Mice were first handled and accustomed to head fixation before starting the schedule of water restriction. Before imaging began mice were conditioned to lick for reward after the go sound (presented within a similar trial structure as the task itself). Imaging began only after mice reliably licked for the presented sound (typically after the 1st day; 200–400 trials). On the first day of imaging, mice were presented with the go sound for 50 consecutive trials, after which the no-go sound was gradually introduced (starting from 10% and increasing by 10% approximately every 50 trials). By the end of the 1st day, the no-go sound reached 50% probability (*Guo et al., 2014*). During the 2nd day, most mice continuously licked for both sounds. Thus, after roughly 100 trials, we increased no-go probability to 80% and waited for mice to perform three consecutive CR trials before returning to 50% probability. This was done for several times until mice increased their performance, specifically learning to withhold licking for the no-go sound. In mice that still continued to lick for both sounds we also repeated the no-go sound several times until the mouse performed correctly. In all mice, a 50–50% protocol was reached typically on the 1st or 2nd day. Most mice learned the task within 3–7 days corresponding to roughly 1200–2000 trials (*Figure 1D*). An effort was made to maintain a constant position of the mouse, speaker and cameras across imaging days in order to maintain similar stimulation and imaging conditions.

## Optical fiber setup

As mice trained on and performed the task we continuously imaged MGB population responses. A ferrule patch cable (M79L01; Thorlabs) was connected to the implanted cannula using a mating sleeve (ADAF1; Thorlabs) and the cable end was connected (SMA connector) to an optical imaging setup (FOM-2; MCI), enabling us to excite the MGB at 470 nm (blue; 0.3 mW power output kept constant throughout the experiments) and collect emission light at 520 nm (green). As a control, we also imaged mice during the task while exciting the MGB with a wavelength that does not excite the calcium indicator (565 nm).

## Histology

Mice were given an overdose of Pental and were perfused transcardially with phosphate-buffered saline (PBS) followed by 4% paraformaldehyde (PFA) in PBS. Brains were post-fixed for 12–24 hr in 4% PFA in PBS and then cryoprotected for >24 hr in 30% sucrose in PBS. Then, 100 μm coronal slices of the entire brain were made using a freezing microtome (Leica SM 2000R), incubated for 15 min in 2.5 μg/ml of DAPI (4',6-diamidino-2-phenylindole), mounted onto glass slides, and imaged using an Olympus IX-81 epi-fluorescent microscope with a 4 × and 10 × objective lens (0.16 and 0.3 NA; Olympus). Fiber tracks and GCaMP6f expression were detected in all mice and were mainly localized to the MGB (*Figure 1C*).

## Body tracking

In addition to fiber photometry, we tracked body movements of the mouse during the task (*Figures 1B* and *5A*). The mouse was illuminated with a 940 nm infra-red LED. A body camera monitored the movements of the mouse at 30 Hz (The Imaging Source; DMK 23UV024 camera). We used

movements of both forelimbs, neck and jaw regions to assess body movements (*Figure 1A*; see *Data Analysis* below). Mice performed the task in the dark.

## Data analysis

Data analyses and statistics were performed using custom-written code in MATLAB. For MGB signals, each trial was normalized to baseline several frames before the visual cue (frame 0 division). We did not find a substantial difference in baseline fluorescence across imaging days and also within an imaging day, implying there was no over bleaching of the signal. This enabled us to quantitatively compare ΔF/F signals across long time periods. To better control for possible changes in MGB signal across days, we tested for frequency tuning (playing a range of frequencies 4–40 kHz; 10, 20, 30, 40 dB attenuations relative to 62 dB; duration 0.1 s) in each mouse, both before and after training. We did not see a significant difference in ΔF/F response between the cases (*Figure 4—figure supplement 1*; $p > 0.05$; n=6 mice; Signed rank test).

Next, we divide trials based either on stimuli (i.e. go or no-go) or on choice (i.e. lick or no-lick per stimulus type). MGB ΔF/F signals were plotted in two dimensional temporal spaces where the x-axis is the trial temporal structure, and the y-axis is the learning profile across trials and days (*Figure 2A*). From this 2D temporal space we averaged across trials during the novice and expert phases (defined as the first and last 500 trials respectively; *Figure 2B*). Alternatively, we averaged across time frames within the trial structure, to obtain an MGB response curve across learning (*Figure 4A*; see below).

## Discrimination power between different trial types

To measure how well could neuronal populations discriminate between trial types (i.e. hit, miss, CR and FA), we calculated a receiver operating characteristics (ROC) curve between the distribution of a pair of trial types and calculated its area under the curve (AUC). We define two different AUC measures by comparing hit trials to either FA (stim AUC; magenta; i.e. different stimuli but similar choice) or to Miss (choice AUC; green; i.e. similar stimuli but different choice) trials (*Figure 3A*). The AUC value ranges from 0 to 1 and quantifies the accuracy of an ideal observer. AUC values close to 0.5 indicate low discrimination whereas values near away from 0.5 indicate high discrimination. In addition, we calculated AUC values for each time frame along the trial for the novice and expert conditions separately (i.e. first and last 500 trials). To assess significance, we calculated the sample distribution by trial shuffling between go and no-go sounds (n = 100 iterations). When the signal exceeded the mean ±3 std of the sample distribution it was defined as significant (*Figure 3B*, arrows).

## Calculation of learning curve and MGB response curves

To calculate the learning curve for each mouse, trials were binned (n = 50 trials with no overlap) across learning and the performance (defined as d'=$Z$(hit/(hit+miss)) – $Z$(FA/(FA+CR)) where $Z$ denotes the inverse of the cumulative distribution function) was calculated for each bin. Next, each behavioral learning curve was fitted with a sigmoid function

$$S(t) = a \frac{1}{1 + e^{\frac{-(t-b)}{c}}} \tag{1}$$

Where *a* denotes the amplitude, *b* the time point (in trial numbers) of the inflection point, and *c* the steepness of the sigmoid. A d'=1 was defined as the performance threshold and mice were ordered based on the trial number in which they crossed threshold (i.e. learning threshold; *Figure 1E*). Different pthresholds did not change the order with which mice learned (see *Figure 1E*).

To compare the behavioral learning curve with responses in the MGB we calculated the mean MGB response across learning (in the same 50 trial bins as the learning curve), averaged during late times (0.6–1 s after stimulus onset; 0.2–0.5 s for m2). Our main focus in this study was on the go sound (hit and miss trials grouped together). Therefore, stimulus identity was kept similar across learning. MGB response curves were also fitted with a sigmoid function (black curves in *Figure 4A*; Curves were smoothed with a Gaussian kernel (2σ = 9) for visualization only). The sigmoid fits of the MGB response curves were normalized between 0 and 1 in order to compare between curves from different mice. In addition, we calculated the MGB response curve for each time frame separately. This was correlated with the fixed learning curve of the mouse, to obtain a time course (within the

trial) correlation coefficient value between the MGB response curve and the learning curve (*Figure 4B*; no smoothing applied).

## Calculating body movements

We used a body camera to detect general movements of the mouse (30 Hz frame rate; *Figures 1A* and *5A*). For each imaging day, we first outlined the forelimbs, neck and mouth areas (one area of interest for each), which were reliable areas to detect general movements. Next, we calculated the body movement (one minus frame-to-frame correlation) within these areas as a function of time for each trial. Thresholding at 3 s.d. (across time frames before stimulus cue) above baseline resulted in a binary movement vector (either 'moving' or 'quiet') for each trial (*Gilad et al., 2018*; *Gilad and Helmchen, 2020*). This movement analysis was very strict, where any movement of the forelimb, neck or jaw would be flagged as a period of movement. A detailed movement analysis of each body part is presented in *Figure 5—figure supplement 2*. In addition, the licking onset was always later compared to the movement onset, since one of the detection areas was the jaw area. This was done for each trial to achieve a 2D space of movement probability within the trial temporal structure (x-axis) versus the learning process (i.e. trial number; y-axis; *Figure 5A*). To obtain MGB signals that are 'movement-free', that is do not contain direct effect of body movements, we detected the first movement onset for each trial, defined as 0.2 s before crossing the movement threshold (*Figure 5E*). Next, MGB signals were truncated from movement onset and onwards in a single trial manner. This analysis resulted in MGB trials, each with a different length, that did not contain body movements (*Figure 5F,G*). Similar results were obtained using other body parts such as the nose and mouth (including licking).

## Statistical analysis

In general, non-parametric two-tailed statistical tests were used. The Whitney rank sum test was used to compare between two medians from two populations and the Wilcoxon signed rank test was used to compare a population's median to zero (or between two paired populations). Multiple group correction was used when comparing between more than two groups. Significance was set at p=0.05.

## Acknowledgements

We thank Joseph Jubran for help with the programming of the behavioral system. We thank members of the Mizrahi laboratory for commenting on early versions of this manuscript. This work was supported by an ERC consolidators grant to AM (#616063), Israeli Science Foundation grants to AM (#224/17, #2453/18), and the European Union's Horizon 2020 research and innovation programme under the Marie Skłodowska-Curie grant agreement No 659719 to AG.

## Additional information

### Funding

| Funder | Grant reference number | Author |
| --- | --- | --- |
| ERC consolidator grant | #616063 | Adi Mizrahi |
| Israel Science Foundation | #224/17 | Adi Mizrahi |
| H2020 Marie Skłodowska-Curie Actions | 659719 postdoctoral fellowship | Ariel Gilad |
| Israel Science Foundation | #2453/18 | Adi Mizrahi |

The funders had no role in study design, data collection and interpretation, or the decision to submit the work for publication.

### Author contributions

Ariel Gilad, Conceptualization, Data curation, Formal analysis, Funding acquisition, Investigation, Visualization, Methodology, Writing - original draft, Writing - review and editing; Ido Maor,

Software, Methodology, Writing - review and editing; Adi Mizrahi, Conceptualization, Supervision, Funding acquisition, Investigation, Visualization, Writing - original draft, Writing - review and editing

#### Author ORCIDs
Ariel Gilad  https://orcid.org/0000-0001-8802-8611
Adi Mizrahi  http://orcid.org/0000-0002-1743-6754

#### Ethics
Animal experimentation: All experiments were approved by Institutional Animal Care and Use Committee (IACUC) at the Hebrew University of Jerusalem, Israel (Permit Number: NS-19-15706-4).

#### Decision letter and Author response
Decision letter https://doi.org/10.7554/eLife.56307.sa1
Author response https://doi.org/10.7554/eLife.56307.sa2

## Additional files

### Supplementary files
• Transparent reporting form

### Data availability
The data and custom code that support the findings of this study are publicly available at: https://osf.io/mt3bc.

The following dataset was generated:

| Author(s) | Year | Dataset title | Dataset URL | Database and Identifier |
|---|---|---|---|---|
| Gilad A | 2020 | Auditory thalamus - fiber photometry | https://osf.io/mt3bc | Open Science Framework, 10.17605/osf.io/mt3bc |

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
