## [Decision Letter]

**Acceptance summary:**

This longitudinal photometric study elegantly shows the existence of choice-related signals in the mouse auditory thalamus which develop together with learning. This indicates that choice signals not only reflect the decision of the animal but also the learnt association between sounds and actions. This is a highly interesting conclusion corroborated by the striking observation that the direction of choice signals depends on sound tuning.

**Decision letter after peer review:**

Thank you for submitting your article "Learning-related population dynamics in the auditory thalamus" for consideration by *eLife*. Your article has been reviewed by three peer reviewers, including Brice Bathellier as the Reviewing Editor and Reviewer #1, and the evaluation has been overseen by Andrew King as the Senior Editor. The following individuals involved in review of your submission have agreed to reveal their identity: Tania Rinaldi Barkat (Reviewer #2); Michael Pecka (Reviewer #3).

The reviewers have discussed the reviews with one another and the Reviewing Editor has drafted this decision to help you prepare a revised submission.

Summary:

Using fiber photometry in the medial geniculate body, the authors follow the activity of the non-lemniscal auditory thalamus throughout learning of a Go/No-Go discrimination task using measurements of GCaM6f fluorescence. They find that (i) thalamic responses are affected by choice and that this cannot only be explained by increased movements, (ii) that choice effects are greater after learning and follow the learning time course, (iii) that choice effects are of opposite signs in regions tuned or non-tuned to the Go stimulus.

Result (i) is not novel as it was known that neurons in the thalamus show choice-related activity. Thus, the main novelty lies in results (ii) and (iii), which are very interesting. However, some concerns about these results need to be addressed by new analyses, to secure and strengthen the conclusions.

Essential revisions:

1) The authors provide an interesting analysis of the movements of the animal, showing that movements increase when the animal learns the task. The authors show that compensating for movements maintains the modulation of activity in the thalamus by choice. However, the effect is weakened in the absence of movements. Given that naive mice show less movements and also weaker choice-related activity, it is critical to show that increased modulation by choice in expert mice is not just due to movement. Hence, the learning analysis should be re-done with the same movement correction.

2) It is disappointing that there is much less choice modulation of thalamic activity on No-Go trials (Figure 3—figure supplement 2). Maybe this is because there is less movement of the animals on No-Go trials. Related to the last point, it is would be good to show that even if the choice-related effects are smaller, they are also learnt. Also, the data in Figure 3—figure supplement 2 is presented in AUC. Please also show it in a format closer to the raw data (i.e. DF/F, as in Figure 2 for hit and miss trials).

3) The latency of choice-related signals should be compared to discrimination latencies in behavior (i.e. time at which the Go and No-Go stimuli generate different licking distributions). This would help to show whether this signal could be causal to choice or just a posterior signal. Also some supplementary examples show choice signals that drop below significance again before the end of the sound presentation, so one wonders whether this signal change was represented in the behavior of the mice (longer response latency etc)?

4) It is not really clear in which regions of the auditory thalamus the recordings were made. Based on histology (Figure 1C), it seems that most recordings are from the medial or the dorsal part of MGB. However, the authors also claim that, based on the tuning profiles, the recordings might be from the ventral part (subsection “Calcium imaging from the MGB along learning”). As the ventral MGB is part of the lemniscal pathway, whereas the dorsal and medial MGB are part of the non-lemniscal pathway, knowing where the activity comes from changes the way the data have to be interpreted. Could it also be that the signal is contaminated by inputs coming from other areas, such as the cortex? It is clear how difficult it can be to distinguish the different subdivisions of the MGB, and that additional experiments (for example by silencing cortex to remove cortical feedback in the recorded responses) to clearly answer this would be beyond the scope of this paper. However, the text should be revised to address this more clearly and to avoid confusion.

5) Related to the previous comment – it is surprising that many tuning curves show best frequencies around 10kHz. Could that be a hint of where the recordings were obtained, or where the activity comes from? If the recordings were made at such diverse locations as indicated in Figure 1C, wouldn't you have expected more diversity in the tuning curves (Figure 1—figure supplement 2, Figure 4—figure supplement 1 and Figure 6)?

6) The question of movement-related activity is very interesting (Figure 5). However, the definition of movement onset is not convincing. To our understanding, it is based on the detection of forelimb and neck muscle movements. However, many other movements could be happening earlier. How does this movement onset align, for example, to lick onset (which has been shown in previous studies to happen 200-300ms after tone onset, much earlier than the movement onset shown in Figure 5E)? What about the contribution of movement planning? How does the movement before sound presentation compare in naïve vs. expert mice?

7) The results from Figure 6 are intriguing. How can you explain that the results are frequency dependent? And if it is only neurons with the BF at the GO tone that increase their activity upon learning, whereas the neurons with a BF different from the GO frequency decrease their activity, how would this look at the whole MGB population level?

8) The general introduction is a bit confusing regarding the overall scope/relevance of the study. The functional dichotomy that the authors describe in the Abstract and Introduction to exist between cortex and thalamus seems to be oversimplified / overstated. The field today largely regards the recurrent networks between thalamus and cortex as a functional unit, and that the idea of a simple relay station for the thalamus is outdated. Likewise, the authors correctly note that findings of choice coding like the one presented here were already made in cortex and even in MGB. Hence, as discussed also by the authors, the strong inter-connectivity of MGB with both cortex and midbrain (where task-specific modulation of responses can already be found) make it not very surprising that MGB is involved as well. Yet the Introduction largely lacks an explanation of this high degree of recurrent connectivity and questions "whether MGB encodes learning related modulation beyond the fear system". The authors should provide a more accurate and nuanced description of what is known and what to expect.

Revisions expected in follow-up work:

Related to point 8 above, the novelty in the presented data is not that one can observe learning-related modulation in MGB per se, but the rather direct demonstration of how it might contribute to learning associated plasticity. To further reinforce this point, two questions should be addressed:

1) Can the same phenomenon be observed in A1 or associated cortices of the non-lemniscal pathway, and if so, where do they arise first? The authors do discuss this issue; however one remains wondering if (or to what extent) the reported findings are reflections of cortical processing, so if available, some data on this question would be very useful.

2) How do performance and MGB responses develop a few weeks after mice have learned the task but were not further trained, i.e. to what extent have performance and neural coding of choice crystallized and thus reflect a form of long-term plasticity associated with learning?

---

## [Author Response]

Essential revisions:1) The authors provide an interesting analysis of the movements of the animal, showing that movements increase when the animal learns the task. The authors show that compensating for movements maintains the modulation of activity in the thalamus by choice. However, the effect is weakened in the absence of movements. Given that naive mice show less movements and also weaker choice-related activity, it is critical to show that increased modulation by choice in expert mice is not just due to movement. Hence, the learning analysis should be re-done with the same movement correction.

In the original draft we have already shown that movement-free MGB signal encodes the choice of the mouse (former Supplementary Figure 6, currently Figure 5—figure supplement 1). Following this comment, we have now re-done the learning analysis using the movement-free signals, showing that MGB enhancement is maintained. The results have been added as new Figure 5—figure supplement 1B, C; subsection “Body movements affect MGB responses”.

2) It is disappointing that there is much less choice modulation of thalamic activity on No-Go trials (Figure 3—figure supplement 2). Maybe this is because there is less movement of the animals on No-Go trials. Related to the last point, it is would be good to show that even if the choice-related effects are smaller, they are also learnt. Also, the data in Figure 3—figure supplement 2 is presented in AUC. Please also show it in a format closer to the raw data (i.e. DF/F, as in Figure 2 for hit and miss trials).

We have now added the MGB responses (ΔF/F) for all six mice for all trial types (see new Figure 2—figure supplement 1). Figure 3—figure supplements 1 and 2 show the AUC values for all response types. In fact, we don’t find this result surprising mainly due to the fact that our recording site is, by design, strong to the ‘go’ sound. Responses to the no-go sound are either weak or non-existent. In mice with a response to the no-go sound (mice #4 and #5), choice encoding did indeed develop with learning. In terms of movement, mice move more in FA compared to CR trials, similar to movement differences in Hit vs. Miss, indicating that movement per se cannot fully explain the weak choice encoding in the No-go trials. We refer to this in the main text (see subsection “MGB encodes sounds early and choices late”).

3) The latency of choice-related signals should be compared to discrimination latencies in behavior (i.e. time at which the Go and No-Go stimuli generate different licking distributions). This would help to show whether this signal could be causal to choice or just a posterior signal. Also some supplementary examples show choice signals that drop below significance again before the end of the sound presentation, so one wonders whether this signal change was represented in the behavior of the mice (longer response latency etc)?

Done. We find that the onset of licking was significantly later than the onset of choice encoding, implying a possible causal relationship. We added it only as text (see subsection “MGB encodes sounds early and choices late”). An example from one mouse (mouse #1) is shown in Author response image 1. In case the reviewers think it is critical to add another supplementary figure with this kind of data we will gladly do it.

**Author response image 1. sa2fig1:** 

4) It is not really clear in which regions of the auditory thalamus the recordings were made. Based on histology (Figure 1C), it seems that most recordings are from the medial or the dorsal part of MGB. However, the authors also claim that, based on the tuning profiles, the recordings might be from the ventral part (subsection “Calcium imaging from the MGB along learning”). As the ventral MGB is part of the lemniscal pathway, whereas the dorsal and medial MGB are part of the non-lemniscal pathway, knowing where the activity comes from changes the way the data have to be interpreted. Could it also be that the signal is contaminated by inputs coming from other areas, such as the cortex? It is clear how difficult it can be to distinguish the different subdivisions of the MGB, and that additional experiments (for example by silencing cortex to remove cortical feedback in the recorded responses) to clearly answer this would be beyond the scope of this paper. However, the text should be revised to address this more clearly and to avoid confusion.5) Related to the previous comment – it is surprising that many tuning curves show best frequencies around 10kHz. Could that be a hint of where the recordings were obtained, or where the activity comes from? If the recordings were made at such diverse locations as indicated in Figure 1C, wouldn't you have expected more diversity in the tuning curves (Figure 1—figure supplement 2, Figure 4—figure supplement 1 and Figure 6)?

Indeed, the limited resolution of fiber photometry in combination with bulk viral injections precludes a decisive determination of the exact source of the signal. The histology shows that we were recording from the non-lemniscal pathway, and closer examination of the full FRAs, shows that we picked up signal ranging 1.4-3.8 octaves across mice. Thus, the tuning curves that we show are slightly biased towards highly tuned responses, while responses are in fact broader. That said, we cannot exclude contribution of feedback or even some leak from MGBv, though we think the latter is less likely. We now refer to this issue in the text explicitly (see subsection “Calcium imaging from the MGB along learning using fiber photometry”).

6) The question of movement-related activity is very interesting (Figure 5). However, the definition of movement onset is not convincing. To our understanding, it is based on the detection of forelimb and neck muscle movements. However, many other movements could be happening earlier. How does this movement onset align, for example, to lick onset (which has been shown in previous studies to happen 200-300ms after tone onset, much earlier than the movement onset shown in Figure 5E)? What about the contribution of movement planning? How does the movement before sound presentation compare in naïve vs. expert mice?

The reviewer is correct that licking started several hundred ms after tone onset. Our movement analysis is actually strict and more stringent than common analyses of licking onset. Any movement of the forelimb, neck or jaw is flagged as a period of movement. To explain this analysis further, we now added an additional supplementary figure with new analysis that we hope clarifies this further (see new Figure 5—figure supplement 2, and subsection “Calculating body movements”). New Figure 5—figure supplement 2 show a more systematic view of our analysis which is, again, more stringent as compared to a standard licking analysis (see also point 3 above). We prefer to leave the examples in Figure 5E as they show the diversity of movement and exemplify the sensitivity of our measurement.

Pre-stimulus movement was generally low. In a new analysis, we did not detect much movement in expert mice during the pre-stimulus period (-1 sec to stimulus onset), which was significantly weaker than in naives. This is now clarified in the text (see subsection “Body movements affect MGB responses”).

7) The results from Figure 6 are intriguing. How can you explain that the results are frequency dependent? And if it is only neurons with the BF at the GO tone that increase their activity upon learning, whereas the neurons with a BF different from the GO frequency decrease their activity, how would this look at the whole MGB population level?

We agree that this result is intriguing. Since the number of mice here is small (n=3), it primarily calls for attending to this issue in future experiments rather than being at the core of the paper. Note that since fiber photometry’s signal arises from many neurons it’s impossible to tease apart single-neuron BF responses and we can only speculate about the type of change the MGB undergoes as a whole (see also point 4 and 5 above). Our interpretation is intended to stay at face value. That is, the total response output of the MGB as a whole remains constant whereas go- and no-go tuned neurons are selectively increased or decreased, respectively. Mechanistic explanation to such phenomena is beyond the scope of this paper, but have been suggested elsewhere, and included in the third paragraph of the Discussion.

8) The general introduction is a bit confusing regarding the overall scope/relevance of the study. The functional dichotomy that the authors describe in the Abstract and Introduction to exist between cortex and thalamus seems to be oversimplified / overstated. The field today largely regards the recurrent networks between thalamus and cortex as a functional unit, and that the idea of a simple relay station for the thalamus is outdated. Likewise, the authors correctly note that findings of choice coding like the one presented here were already made in cortex and even in MGB. Hence, as discussed also by the authors, the strong inter-connectivity of MGB with both cortex and midbrain (where task-specific modulation of responses can already be found) make it not very surprising that MGB is involved as well. Yet the Introduction largely lacks an explanation of this high degree of recurrent connectivity and questions "whether MGB encodes learning related modulation beyond the fear system". The authors should provide a more accurate and nuanced description of what is known and what to expect.

Thanks for pointing this out. We agree and now restructured the Introduction such that learning-related changes are at the core of the text. We also made some changes in the Discussion.

Revisions expected in follow-up work:Related to point 8 above, the novelty in the presented data is not that one can observe learning-related modulation in MGB per se, but the rather direct demonstration of how it might contribute to learning associated plasticity. To further reinforce this point, two questions should be addressed:1) Can the same phenomenon be observed in A1 or associated cortices of the non-lemniscal pathway, and if so, where do they arise first? The authors do discuss this issue; however one remains wondering if (or to what extent) the reported findings are reflections of cortical processing, so if available, some data on this question would be very useful.

Given that the main effect is in a late time during the trial, we expect the changes are indeed of a non-lemniscal source. As for the origin of this phenomenon, we currently have no data on this issue. As we discuss, electrophysiology (preferably in multiple brain regions) of the lemniscal and non-lemniscal pathway simultaneously, could be a great follow-up work to answer this.

2) How do performance and MGB responses develop a few weeks after mice have learned the task but were not further trained, i.e. to what extent have performance and neural coding of choice crystallized and thus reflect a form of long-term plasticity associated with learning?

Unfortunately, we do not have data on mice that learned but not further trained and then imaged again. We have data on mice that continued to train on harder tasks, but these had unforeseen issues, like instability of behavioral performance at perceptual limits, and technical issues developing after months of imaging. We suspect that the strong choice signal in MGB is malleable and would change with different behavioral and cognitive demands, but we have no data to support this.